# ACPA-Net: Atrous Channel Pyramid Attention Network for Segmentation of Leakage in Rail Tunnel Linings

**Peng Geng [1], Ziye Tan [1], Jun Luo [1], Tongming Wang [2,*], Feng Li [3,4,*] and Jianghui Bei [5]**

[1] School of Information and Science Technology, Shijiazhuang Tiedao University, Shijiazhuang 050043, China
[2] Department of Computing, Hengshui University, Hengshui 053000, China
[3] College of Civil and Transportation Engineering, Shenzhen University, Shenzhen 518060, China
[4] Institute of Intelligent Transportation and Safety Operation, Shenzhen University, Shenzhen 518060, China
[5] Highway Safety Perception and Monitoring Key Laboratory of Hebei Province, Shijiazhuang 050043, China
* Correspondence: hsncwtm@163.com (T.W.); lf260986@semi.ac.cn (F.L.)

**Abstract:** The automatic segmentation of leakage in rail tunnel linings is a useful and challenging task. Unlike other scenarios, the complex environment inside the tunnels makes it difficult to obtain accurate results for the segmentation of leakages. Some deep learning-based methods have been used to automatically segment tunnel leakage, but these methods ignore the interdependencies between feature channels, which are very important for extracting robust leakage feature representations. In this work, we propose an atrous channel pyramid attention network (ACPA-Net) for rail tunnel lining leakage segmentation. In ACPA-Net, the proposed atrous channel pyramid attention (ACPA) module is added into a U-shaped segmentation network. The ACPA module can effectively strengthen the representation ability of ACPA-Net by explicitly modeling the dependencies between feature channels. In addition, a deep supervision strategy that helps ACPA-Net improve its discrimination ability has also been introduced into ACPA-Net. A rail tunnel leakage image dataset consisting of 1370 images with manual annotation maps is built to verify the effectiveness of ACPA-Net. The final experiment shows that ACPA-Net achieves state-of-the-art performance on the Crack500 dataset and our rail tunnel leakage image dataset, and our method has the least number of parameters of all the methods.

**Keywords:** channel attention; convolutional neural network; deep supervision; leakage segmentation; rail tunnel lining

## 1. Introduction

A tunnel is an important structure of a rail transit system. Due to the influence of the surrounding environment, cracks and leakages are commonly observed on the surface of rail tunnel linings [1,2]. Leakage and cracks are two of the main tunnel defects, accounting for 84% of the total number of defects in rail tunnels [3]. Tunnel lining leakage is a disaster that is easy to be ignored in the early stage, but as time passes, it may affect the stability of the tunnel to some extent. Therefore, it is particularly important to detect the leakage of the tunnel in time. In addition to effectively guaranteeing traffic safety, the timely detection of leakages can also reduce the costs of subsequent maintenance. Traditional leakage detection mostly relies on manual detection. The inspectors walk along the inside of the tunnel with lighting equipment. Such traditional leakage detection methods haves many disadvantages. The use of manual detection consumes considerable time and material resources. Moreover, it is difficult for people to provide objective and quantitative information due to the influence of subjective factors. Therefore, it is of great significance to develop a rapid, accurate, and comprehensive method for detecting tunnel leakages. It is a better choice to use an image recognition algorithm to analyze tunnel lining leakage.

In recent years, some deep learning-based image segmentation methods have been used for tunnel leakage detection. These methods can be classified into two-stage and one-

stage image segmentation networks according to whether they are based on the existence of an object detection network. Two-stage methods detect the leakage areas or collect some region proposals through the object detection network and then send these areas to the fully convolution neural network for subsequent segmentation. Wu et al. [4] and Zhao et al. [5] directly used Mask R-CNN to detect tunnel leakage regions. Inspired by Mask-RCNN [6], Gao et al. [7] proposed an FCN-RCNN model by combining Faster-RCNN with FCN [8]. Zhao et al. [9] replaced the backbone network of Mask-RCNN with ResNet-101 for tunnel leakage instance segmentation. Xue et al. [10] combined an object detection network, named the single shot multibox detector (SSD), with the fully convolution network FCN [8] into a new network [10]. Because the one-stage object detection network SSD has a faster inference speed than the two-stage object detection network of the RCNN series, the SSD-FCN can segment the leakage areas faster than the previous Mask-RCNN. In the same year, Xue et al. [11] applied the cascade Mask-RCNN proposed by Cai et al. [12] to tunnel leakage image segmentation. The two-stage methods above have better segmentation performance than the one-stage methods, but the two-stage methods have heavy computational complexity and a large memory footprint, resulting in slow inference speed. Moreover, the performance of the two-stage methods above heavily depends on the quality of the bounding-box proposals generated by the object detection network in two-stage methods. The environment in the tunnel is complex. The cables, pipes, stains, shadows, and manual traces on the lining surface interfere with the object detection network to produce high-quality proposals which will seriously affect the subsequent segmentation.

The one-stage image segmentation method means that images are directly sent into the convolution neural network to extract features, and the final result is predicted directly from them. These methods are simpler and faster than two-stage segmentation methods. One-stage image segmentation methods are also the most commonly used in image segmentation. Based on FCN, Huang et al. [1] proposed a two-stream algorithm for the simultaneous segmentation of cracks and leakages. Xiong et al. [13] divided tunnel leakage images into six categories using FCN network. Cheng et al. [14] used the improved FCN to segment the tunnel leakage intensity image automatically, which is a new method that uses the intensity images of terrestrial mobile LiDAR (Light Detection and Ranging) for automatic leakage detection in shield tunnels based on deep learning; this method will be abbreviated as FCN(VGG19) in the following paragraphs. In the FCN-based methods, the larger ratio upsampling operation makes the predicted segmentation result fuzzier and more insensitive to the details in the image. So, the segmentation result is not fine enough. The DeepCrack [15] method is used to fuse multi-scale deep convolution features extracted from the hierarchical convolution stages with VGG. Similar to DeepCrack, [16] multi-scale levels of densely connected layers instead of VGG are used to extract significant features of crack images. AugMoCrack [17] inputs the concatenated feature map into the cross-stage partial (CSP) and convolution layers to obtain multi-scale detection feature maps, and then, it uses a confidence comparison to determine the final prediction with the highest confidence. The CSP layer decreases the number of inference computations by dividing the gradient flow and increases the detection performance. Compared with the FCN-based method, these methods fusing different stages of feature maps make prediction results more accurate. Furthermore, UNet-based methods [18] use an encoder–decoder structure and the skip connection strategy to concatenate the output of each encoding block with the feature map of the decoding block of the same level to recover the semantic information lost during the encoding process. UNet-based methods can more perfectly recover the details in the original image, and they greatly improve the segmentation performance. Huanyan et al. [19] proposed a kind of UNet structure constructed by residual blocks to segment cracks in asphalt concrete pavement images. In order to deal with the problem that continuous cracks are interrupted in segmentation, a skip-level round trip sampling block [20] is proposed to fuse multi-scale pooling (or up-sampling) to combine the characteristics of different receptive fields based on a UNet prototype. These UNet-based methods

use an encoder–decoder structure and the skip connection strategy to concatenate the shallow feature maps with the deeper feature maps in order to more perfectly recover the details in the original image, and they greatly improve the segmentation performance. In U-shaped networks, not all the features obtained by the encoder are effective for segmentation. The normal convolution operation equally extracts the features in each channel, which is different from the attention module taking into account the most informative feature representations channel-wise. Under weak lighting conditions, the cables, pipes, stains, shadows, and manual traces on the tunnel lining surface interfere with the segmentation of leakage images with those methods.

To address the issues mentioned above, an atrous channel pyramid attention network (ACPA-Net) shown in Figure 1 is proposed for rail tunnel leakage image segmentation. A lightweight channel attention ACPA module is proposed to effectively capture the cross-channel interaction to strengthen useful features, suppress useless features, and extract robust feature representations. Then, the proposed ACPA module is incorporated into a U-shaped encoder–decoder network to construct the proposed ACPA-Net architecture. In addition, a side network is added to the decoder of ACPA-Net to predict the output of each step in the decoder. These predicted outputs are used to further supervise ACPA-Net to achieve more accurate results.

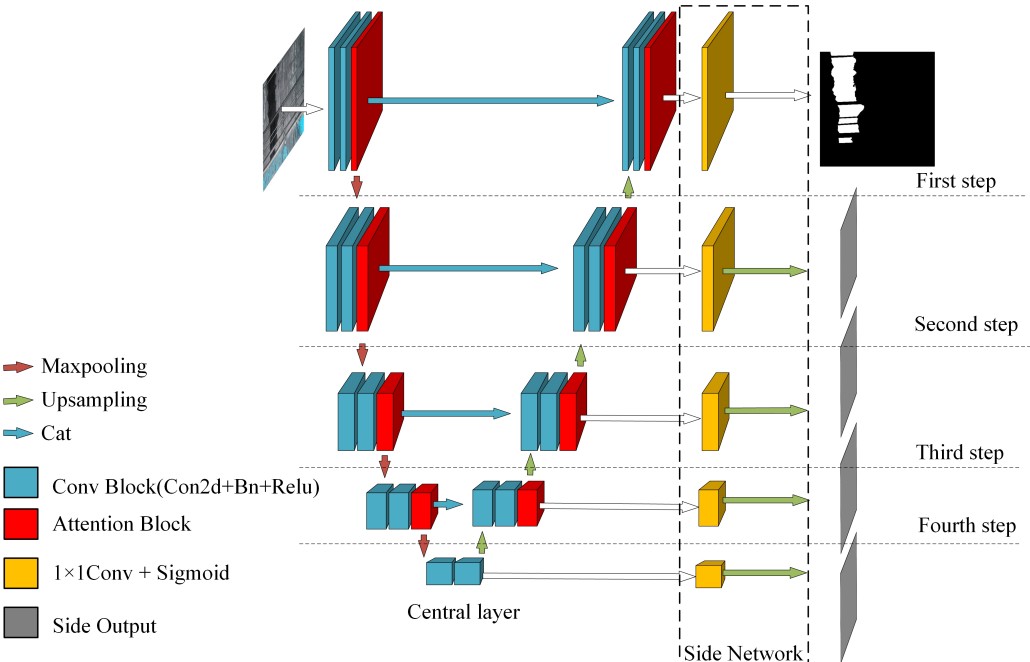

**Figure 1.** The architecture of the atrous channel pyramid attention network (ACPA-Net).

The main contributions of our work are summarized as follows:

1.  We propose an ACPA module, which is a lightweight channel attention module. The ACPA module can effectively capture longer distance channel interaction. The ablation study shows that the ACPA module can help ACPA-Net learn strong semantic feature representations effectively.
2.  An effective ACPA-Net is proposed to segment the leakage regions in the rail tunnel image. ACPA-Net uses a U-shaped network as a basic network structure and combines it with the proposed ACPA module.
3.  The combined model with the deep supervision strategy is applied to the Crack500 [21] dataset and our tunnel leakage dataset. Then, excellent performance is obtained, which proved the effectiveness of the model in rail tunnel leakage segmentation.

## 2. Related Work

In recent years, attention mechanisms have been successfully applied to various computer vision tasks, such as image classification and semantic segmentation. Self-attention [22], which is proposed for machine translation, is thought to be used for the first time to capture the global dependencies of the input feature maps [23,24]. An attention residual learning mechanism [25] is proposed to optimize a very deep residual attention network with hundreds of layers. In SENet [26], a channel-attention network is designed to adaptively predict potential key features by explicitly modeling the interdependence between channels. In other words, by learning the correlation between channels, SENet can pick out the attention specific to the channels and then can learn to use global information to selectively emphasize informative features and suppress less useful ones. Li et al. [27] proposed an SE module with a skip-connection to emphasize useful channels and suppress useless channels of the feature maps in a kind of UNet-shape crack segmentation structure with VGG-16 as Encoder and densely connected structure as Decoder. Based on the SE module, concurrent spatial and channel squeezes and excitation [28] (scSE) combine spatial attention with channel attention in a parallel manner. Stephen et al. [29] introduced an scSE block into UNet-based Pavement Crack Segmentation architecture with a ResNet-34 neural network. On the other hand, the convolutional block attention module [30] (CBAM) combines spatial and channelwise attention in a sequential arrangement. Zhou et al. [31] proposed a tunnel leakage image segmentation method [31] based on DeepLabv3+ with EfficientNet as the backbone. CBAM is inserted between the ASPP structure and $1 \times 1$ convolutions to magnify the weight of effective feature channels in the feature layer so that the model can better distinguish the leakage with background. A nonlocal [32] module is proposed to capture spatial long-distance dependencies, which is based on the weights of the relationship between any pixels in a 2D feature map and the current pixels. The nonlocal module is embedded in a pavement crack network based on the modified ResNet-101 network with atrous convolution [33]. The nonlocal module makes use of the relationship between the local and global context feature, which can increase the relationship weight of road cracks and improve the detection accuracy. DANet [24] introduces a position attention module and a channel attention module to capture global dependencies in the spatial and channel dimensions, respectively. A modified ResNet-based network architecture with a multiscale fusion process was proposed for learning hierarchical features of cracks [34]. The dual attention module was incorporated into this architecture to better separate tiny cracks from the background.

Obviously, all of the above-mentioned attention modules focus on developing sophisticated attention modules for better performance. The attention modules mentioned above increase the complexity of different segmentation networks due to the mass of parameters and their higher model complexity. Hence, Wang et al. [35] proposed an efficient channel attention (ECA) module based on an SE module. In the ECA module, the two-layer full connection in the SE block was replaced by a 1D convolution with a convolution kernel. The number of parameters in the ECA module is greatly reduced. Hence, the ECA module overcomes the trade-off between performance and complexity. It not only has a few parameters but also significantly improves the performance. By the aid of the ECA module, the importance of each feature channel can be represented by modeling, and the neighboring channels are correlated, and the weight of each feature channel will be calculated by its neighbor channels. So, it can avoid dimensional loss while capturing local cross-channel interactive information. However, the ECA module [35] only considers the cross-channel interactions of k adjacent channels, so long-distance channel interaction cannot be obtained. Therefore, by considering more and longer distance cross-channel interactions, the attention weights can be more reliable. Atrous convolution can expand the receptive field without losing resolution and without introducing additional parameters. Thus, we propose the ACPA module, which replaces the common 1D convolution with multiple parallel 1D dilated convolutions with different atrous rates, and we applied the

module to the tunnel leakage segmentation network structure. In the next section, we will introduce the ACPA-Net structure and ACPA module in detail.

## 3. Method

In this section, we first overview the proposed ACPA-Net. Then, we introduce the major setting of the network in detail.

### 3.1. Network Architecture

We formulate tunnel lining leakage segmentation as a pixelwise binary classification task, where leakage areas are marked as 1 and nonleakage areas are marked as 0. When an image with leakage is input into the network, we expect the leakage area in the output of the last layer to have a high probability value, while other places have a low probability value.

Having high-level contextual information along with adequate spatial information is crucial to achieve high-quality segmentation results. To obtain a larger receptive field, the feature maps are usually downsampled; hence, the feature maps with high level representation usually have a low resolution. It means that a large amount of spatial information will be lost. In contrast, low-level feature maps have rich spatial information but lack high-level context information. To solve this problem, we use a UNet-like model as our basic network architecture. Furthermore, the feature maps in the encoder with similar resolution to the decoder are integrated into the decoder through skip connections. By fusing the hierarchical features of the encoder, the decoder gradually increases the spatial resolution and fills the missing details [36].

Figure 1 shows the overall framework of the proposed ACPA-Net. There are 4 steps each in the encoder and decoder, and the encoder and decoder are connected through the central layer. In the encoder, each downsampling step is composed of two convolution blocks: an attention block and a downsampling layer. Each convolution block consists of a $3 \times 3$ convolution, a batch normalization (BN) layer, and a rectified linear unit (ReLU). The attention block uses the ACPA module, whereas a $2 \times 2$ maxpooling layer with a step size of 2 is used in the downsampling layer. Moreover, each upsampling step in the decoder uses the nearest neighbor sampling to double the upsampling first, and the following two convolution blocks and attention modules are the same as the downsampling step in the encoder. In the central layer, there are only two convolution blocks.

### 3.2. Atrous Channel Pyramid Attention

The proposed atrous channel pyramid attention (ACPA) module is illustrated in Figure 2. It is an extension of the ECA module [35]. ECANet focuses on the question of whether effective channel attention mechanisms can be learned in a more efficient way. The idea of channel attention is developed by SENet [26]; in SENet, a channel-attention network is designed to adaptively predict potential key features by explicitly modeling the interdependence between channels [26]. Specifically, given the input features, the SE block first applies global average pooling independently for each channel and then uses two fully connected (FC) layers and a nonlinear sigmoid function to generate channel weights. The two FC layers are designed to reduce the dimensions to control the complexity of the model. Although this strategy can reduce the computation, there is a process in which the channels are compressed. It leads to an indirect correspondence between channels and their weights, which can affect ACPA-Net performance.

Wang et al. [35] proved that the channels and their weights need to be in direct correspondence. This means that avoiding dimensionality reduction is more important than considering nonlinear channel dependencies. Therefore, Wang et al. [35] proposed an efficient channel attention (ECA) block based on the SE block. The two-layer full connection in the SE block was replaced by a 1D convolution with a convolution kernel size of *k*. In this way, the channel with its weight has a direct correspondence, and the number of parameters is greatly reduced. However, the interaction range between channels is *k* adjacent channels. More reliable attention weights can be obtained by considering a wider

range of cross-channel interactions. Inspired by this, we propose an ACPA module, which replaces the common 1D convolution with size k in the ECA module into multiple parallel 1D convolution with different atrous rates. The 1D convolution with size k can only carry out in the ECA module with multiple parallel 1D dilated convolutions with different atrous rates to capture the cross-channel interaction within the multiscale distance.

As shown in Figure 2, given a feature map $F \in \mathbb{R}^{H \times W \times C}$ with height $H$ and width $W$ as input, it can be expressed as $F = [f_1, f_2, f_3, \cdots, f_c]$, which is composed of channels $f_i \in \mathbb{R}^{H \times W}$. Global average pooling (GAP) is used to compress it in the spatial dimension to aggregate spatial information, so the feature vector $F_{GAP} \in \mathbb{R}^{1 \times 1 \times C}$ can be obtained. Its $k^{\text{th}}$ element can be expressed as:

$$f_{GAP(k)} = \frac{1}{H \times W} \sum_{i}^{H} \sum_{j}^{W} f_k(i,j)$$

(1)

where $f_k(i,j)$ denotes the pixel$(i,j)$ in $f_k$. For the convenience of the 1D convolution operation, we remove a dimension of $F_{GAP}$ and obtain $F_{GAP} \in \mathbb{R}^{1 \times C}$.

$$F_{GAP} = \left[ f_{GAP(1)}, f_{GAP(2)}, f_{GAP(3)}, \cdots, f_{GAP(C)} \right].$$

(2)

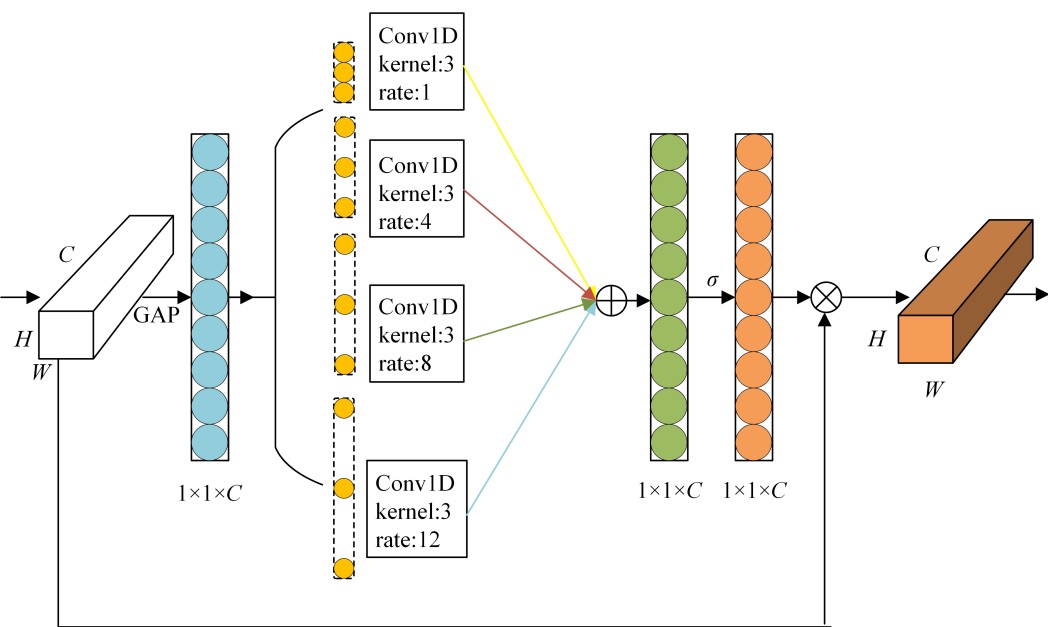

**Figure 2.** Architecture of the ACPA module.

To capture the cross-channel interaction of the feature maps in the multiscale distance, $F_{GAP}$ is input into four parallel 1D atrous convolutions with different atrous $rates = (r_1, r_2, r_3, r_4)$ (e.g., employing $rates = (1, 4, 8, 12)$ for the four parallel branches). Here, 1D convolution is an effective way to make all channels share the same learned parameters. It can efficiently capture the cross-channel interaction of adjacent $K$ feature channels. The four parallel 1D atrous convolutions can expand the cross-channel interaction of capturing $K$ adjacent channels into the cross-channel interaction of channels with multiple scale distances. We obtain the output $Y \in \mathbb{R}^{1 \times C}$ by the elementwise addition of all the outputs of the four parallel atrous 1D convolutions. Therefore, the output $Y[i]$ of the parallel atrous convolutions of an input $F_{GAP}[i]$ with a filter $w[k]$ of length $K$ is defined as:

$$Y[i] = \sum_{j=1}^{4} \sum_{k=1}^{K} F_{GAP}[i + k \cdot \text{rates}[j]] \cdot w[k]$$

(3)

where $rates[j]$ denotes the $j^{\text{th}}$ rate in atrous convolution and the $rates[j] - 1$ zeros are inserted into both sides of the $F_{GAP}$ before the atrous convolutions to keep the output size the same as the input size. Then, we obtain the channel attention map through a Sigmoid function to rescale $Y$ to [0,1]. Accordingly, the operation of the whole ACPA module can be simply described as follows:

$$W = \sigma\left(\sum_{i=0}^{3} C1D_K^{\text{rates}[i]}(GAP(F))\right) \tag{4}$$

where $C1D_K^{\text{rates}[i]}$ refers to the 1D convolution layer with atrous ratio $rates[i]$ and convolution kernel size $K$. $F$ is the input of the ACPA module. $GAP$ and $\sigma$ represent the global average pooling layer and sigmoid layer, respectively. Given the channel attention map $W$, it can be used to emphasize important channels of $F$ and suppress less useful ones:

$$\tilde{F} = ACPA(F) = W \otimes F \tag{5}$$

where $\otimes$ refers to elementwise multiplication between the channel attention map $W$ and the input feature map $F$. During multiplication, the channel attention map $W$ broadcasts along the spatial dimension. The ACPA in (5) indicates the whole atrous channel pyramid attention module. $\tilde{F} \in \mathbb{R}^{H \times W \times C}$ denotes the final output whose channel information is selectively enhanced or suppressed by the ACPA module.

In ACPA-Net, the ACPA module is inserted at the end of the two convolution blocks in each step. Given an input $X$, the output $Y$ of a step can be expressed as:

$$Y = ACPA(ConvB\,(ConvB\,(X))) \tag{6}$$

where $ConvB$ represents the convolution block, which is composed of a $3 \times 3$ convolution, a batch normalization (BN) layer, and a rectified linear unit (ReLU).

### 3.3. Deep Supervision

Deep supervision adds some additional constraints in the front layer of the network. It can increase the convergence speed and improve the accuracy of the network in some cases. Lee et al. [37] first applied deep supervision in the training of deep networks. They proved that the use of deep supervision can improve the discriminativeness and robustness of features learned by the network, especially in the early layers [37]. The effectiveness of deep supervision has also been verified in the edge detection network HED [38] and the medical image semantic segmentation network UNet3+ [39]. Therefore, we use deep supervised learning in ACPA-Net to make our network more discriminative.

To use deep supervision in ACPA-Net, we add the side network into the decoder of the network. The side network is used to predict the output of each level of ACPA-Net for deep supervision learning. The side network is composed of a $1 \times 1$ convolution layer, a nearest neighbor upsampling layer and a sigmoid layer. The output of the central layer and the feature maps generated after each upsampling step in the decoder are used as the input of the side network. First, $1 \times 1$ convolution is used to predict the result and obtain the side output. Then, except for the side output with the original resolution in the first layer, the others will be restored to the same resolution as the input image through the upsampling layer. Finally, five side outputs supervised by the ground truth are obtained by the sigmoid layer.

During the supervision process, the loss function is used to calculate the difference between the side outputs and ground truth. Binary cross-entropy loss (BCE loss) and Dice-coefficient loss (Dice loss) are two commonly used loss functions in image segmentation. These two loss functions are pixel-level and map-level measures, respectively [39]. The Dice coefficient is an important indicator used to measure the quality of segmentation results. Dice loss can directly optimize the network in the direction of the optimal Dice coefficient. As a region-related loss, Dice loss has good performance for scenarios where

the positive and negative samples are severely imbalanced, but the small targets in the training samples can make the loss in the training process unstable. However, BCE loss is more stable than Dice-coefficient loss. To take advantage of both Dice loss and BCE loss, we propose a hybrid loss that combines BCE loss and Dice loss with different weights. The definition of hybrid loss is as follows:

$$l_{\text{seg}} = \alpha l_{\text{bce}} + l_{\text{dice}} \tag{7}$$

where $l_{\text{bce}}$ and $l_{\text{dice}}$ refer to BCE loss and Dice loss, respectively. Here, $\alpha$ is a parameter controlling the weight of BCE loss between the losses, and it is equal to 0.5 in our experiments. We use BCE loss to make all pixels have a smooth gradient and then use Dice loss to make ACPA-Net pay more attention to the foreground [40,41]. Finally, our loss is defined as the weighted sum of the losses of each side output:

$$L = (1/M) \sum_{m=1}^{M} l_{\text{seg}}^{m} \tag{8}$$

where $M$ refers to the total number of side outputs, $l_{\text{seg}}^{m}$ refers to the hybrid loss calculated by the $m^{\text{th}}$ side outputs, and $L$ denotes the final loss of the entire network, which is obtained by averaging the losses of all side outputs $l_{\text{seg}}^{m}$.

## 4. Experiment

### 4.1. Implementation Details

The implementation is based on PyTorch 1.4.0. The operating system is Windows 10. An Intel Core i5-10400F CPU and a GeForce GTX 1660 SUPER GPU are used for both training and testing. All the experiments below are conducted on the same training set and testing set. The training mini-batch size is set to 4. The optimizer uses RMSProp with an initial learning rate of $1 \times 10^{-4}$, momentum of 0.9 and weight decay of $1 \times 10^{-8}$. ReduceLROnPlateau is used as a learning rate adjustment strategy by which the existing learning rate will be multiplied by 0.1 when the validation loss stays the same for 3 epochs. When the learning rate reaches $1 \times 10^{-8}$, there will be no further decline. The early stop mechanism is used in the experiment. That is, when the F1-score of the validation set has not improved within 10 epochs, the training will be terminated earlier. ACPA-Net is initialized by kaiming_normal [42]. In later experiments, we fix the values of the parameters discussed above.

### 4.2. Datasets

Since there are no publicly available datasets for rail tunnel leakage images, with an imaging device, tunnel lining images are collected in the rainy season from the different railway tunnels in the mountain area along the Shuohuang Railway crossing the Taihang Mountains between Hebei Province and Shanxi Province in the north of China. We conduct experiments with the dataset collected by ourselves. In view of the difficulty of obtaining tunnel lining images with water leakage, our research team spent 2 months using several methods to enlarge the number of tunnel lining images. Firstly, because the water leakage area of the tunnel lining will show different shapes and sizes after heavy rain, the same water leakage regions are photographed multiple times especially before heavy rain and after heavy rain. Secondly, the same water leakage area is photographed from different positions and perspectives. When capturing the tunnel lining, the same water leakage area is avoided to completely display it in the different images. A large and complex water leakage region can partially be photographed from different positions and perspectives and displayed in the different images. Finally, the images containing water leakage are selected from the collected tunnel lining images, and the selected images are cropped to highlight the water leakage region. The images and ground truths are resized as $512 \times 512$. The operation of cropping the images into a square shape ensures that the scaling operation will not distort the images. According to the proportion of 4:1, the images are divided into

training dataset and test dataset. The images containing leakage disease are selected and manually marked by LabelMe [43] software. The number of images in the training dataset is 1100, and that in the test dataset is 270. Due to the limitation of GPU memory, all images are scaled to a size of 256 × 256 in the experiment.

As shown in Figure 3, there are some representative images and corresponding annotations from our rail tunnel leakage image dataset. It is observed that the environment inside the tunnel is complex: some leakage areas have simple backgrounds, but some leakage areas are covered by pipes, cables or shadows. Artificial traces, mosses and other disturbances similar to the color of the leakage areas make segmentation difficult. In addition, there are some blurred images and low gray value images in the dataset, which also bring challenges to our water leakage segmentation technique.

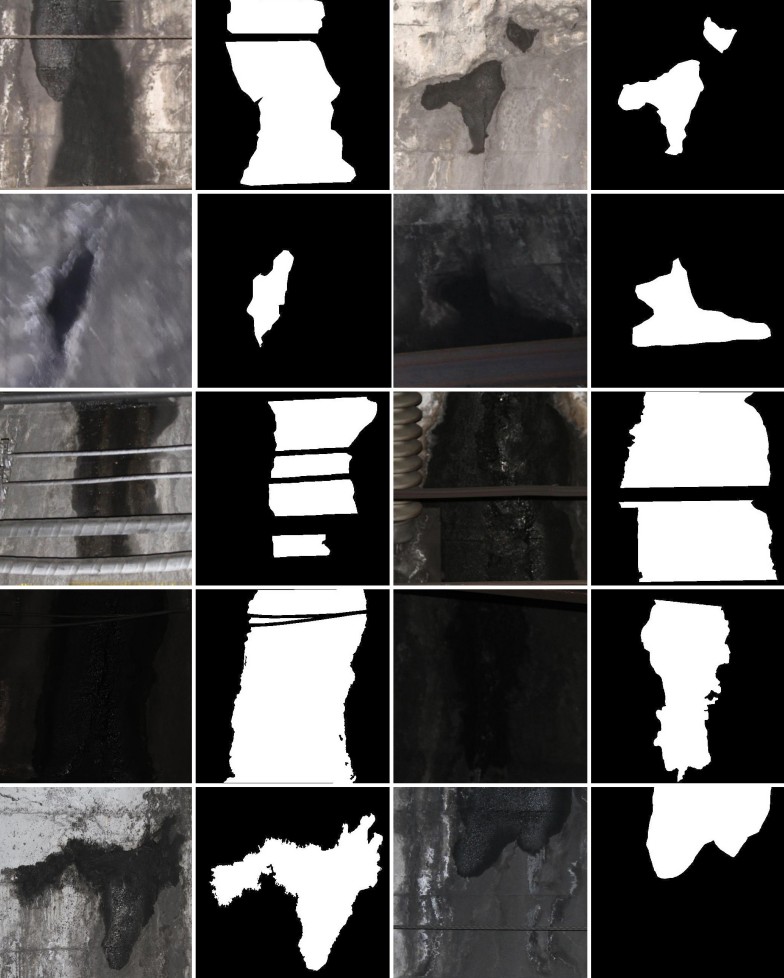

**Figure 3.** Representative images and ground truth of our rail tunnel leakage image dataset.

The performance of our model in the accurate prediction of fine cracks and crack edges is explored using the Crack500 [21] dataset. The Crack500 dataset includes 3368 pavement crack images from 500 raw crack images with around 2000 × 1500 pixels captured by a cell phone. All the raw images are manually labeled at the pixel level. In order to prompt the image processing speed and reduce the consumption of the computation resources, each image is resized to 256 × 256 pixels. In Crack500, 1896 images are used as the training samples, and 348 images are used as the validation samples.

### 4.3. Evaluation Metrics

To evaluate the proposed ACPA-Net, we conduct a quantitative evaluation of the prediction results by the following metrics: Precision, Recall, intersection-over-union (IoU),

F1-score, and Accuracy. Precision indicates how many leakage areas in the output are correctly predicted. Specifically, Precision represents the proportion of pixels that are correctly predicted as leaking areas among all pixels that are predicted to have leakage. Recall indicates how many leakage areas in the label map are correctly predicted. Specifically, it means the proportion of pixels that are correctly predicted as leakage areas among the pixels that actually leak. A higher Precision or a higher Recall does not mean that a network has good performance. Therefore, Precision and Recall are often used together to measure the effectiveness of a network. The F1-score is mathematically the same as the Dice coefficient, which comprehensively considers Precision and Recall. An excellent network needs to find a balance between Precision and Recall, so the F1-score is a reliable indicator for evaluating network performance. IoU is also an important indicator that represents the intersection over union ratio between the leakage areas in the segmentation result and the leakage areas in the ground truth. Accuracy is defined as the percentage of pixels that are correctly predicted, including leakage and background, i.e., all pixels. These metrics are computed as follows:

$$\text{Precision} = \frac{TP}{TP + FP} \tag{9}$$

$$\text{Recall} = \frac{TP}{TP + FN} \tag{10}$$

$$\text{IoU} = \frac{TP}{TP + FP + FN} \tag{11}$$

$$\text{F1-score} = \frac{2 \times \text{Precision} \times \text{Recall}}{\text{Precision} + \text{Recall}}$$
$$= \frac{2TP}{2TP + FP + FN} \tag{12}$$

$$\text{Accuracy} = \frac{TP + FP}{TP + TN + FP + FN} \tag{13}$$

True positive ($TP$) refers to the number of pixels in the segmentation result that are correctly identified as the leakage area. False positive ($FP$) refers to the number of pixels identified as the leakage area in the segmentation result but actually belong to the background area in the label map. True negative ($TN$) means the number of pixels that are correctly identified as the background area. False negative ($FN$) represents the number of pixels identified as the background in the segmentation result but that actually belong to the background leakage area in the label image.

### 4.4. Ablation Studies

#### 4.4.1. The Effect of Hybrid Loss

To prove the effectiveness of the hybrid loss function and analyze the sensitivity of parameter $\alpha$ in the hybrid loss function, we conduct a series of experiments on the proposed network over different loss functions. The loss functions include BCE loss, Dice loss, and hybrid loss functions. Figure 4 shows the curves of the indicators F1-score and Accuracy during training. BCE loss, Dice loss, and hybrid loss are marked as "BCE", "DL", and "$\alpha$BCE+DL" in Figure 4, respectively. The $\alpha$ in "$\alpha$ BCE+DL" is selected as 0.25, 0.5, 1, 1.5, and 2. Figure 4 shows that the training process fluctuates greatly when the Dice loss is used. There is a more stable training process when BCE loss is chosen. Compared with Dice loss, the use of hybrid loss makes the training process more stable. More importantly, compared with Dice loss and BCE loss, the hybrid loss makes ACPA-Net perform better. Note that the closer $\alpha$ is to 1, the better the ACPA-Net performance. This means that when the proportion of BCE loss in the hybrid loss is closer to that of Dice loss, better results can be obtained.

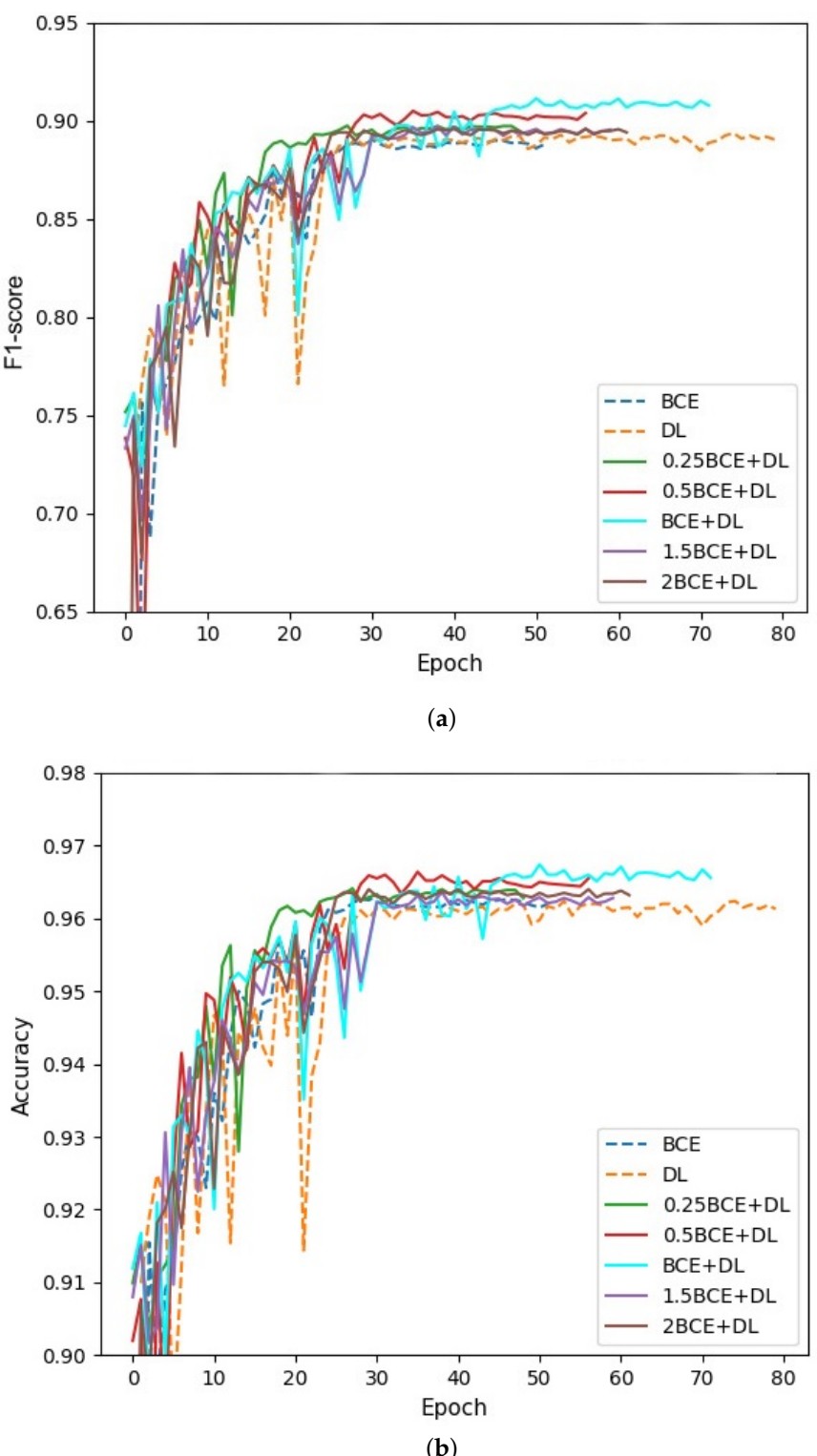

**Figure 4.** Validation performance for different coefficients of hybrid loss. (**a**) Validation F1-score for different coefficients of hybrid loss. (**b**) Validation Accuracy for different coefficients of hybrid loss.

### 4.4.2. The Effect of the ACPA Module and Deep Supervision

To verify the effectiveness of the proposed ACPA module and the deep supervision module for the segmentation of leakages, we take UNet as the baseline and compare its results with those integrating the ACPA module and the deep supervision module. The implicit parameters of all models, such as batch size and learning rate, are set the same

to ensure the fairness of the experiment. The experiment is carried out on our rail tunnel leakage image dataset. The F1-score, IoU, and Accuracy are selected as evaluation metrics. The following parameters in the ACPA modules are used in all of the experiments. We set the atrous rates of the ACPA module to (1, 2, 4, 8) for the first step of the encoder and decoder, and we set the atrous rate of the ACPA module in other steps to (1, 4, 8, 16). The kernel size of the 1D convolution is 3.

The experimental results are shown in Table 1. We first introduce the ACPA module into UNet. Compared with the baseline, the introduction of the ACPA module improve the F1-score, IoU, and Accuracy by 1.55%, 2.1%, and 0.37%, respectively. In particular, when we further add the deep supervision strategy to ACPA-Net, we achieve the best F1-score, IoU, and Accuracy, which are 90.75%, 83.62%, and 96.68%, respectively. Compared with the baseline, the F1-score is improved by 3.9%, and the IoU and Accuracy are also greatly improved.

**Table 1.** Ablation study of each component inserted into the baseline.

| Methods | F1-Score (%) | IoU (%) | Accuracy (%) |
| --- | --- | --- | --- |
| UNet (Baseline) | 86.85 | 77.86 | 95.43 |
| +ACPA | 88.4 (1.55↑ [1]) | 79.96 (2.1↑) | 95.80 (0.37↑) |
| +ACPA+DS | **90.75 (3.9↑)** | **83.62 (5.76↑)** | **96.68 (1.25↑)** |

[1] ↑ means the percentage point improvement in each metric compared to UNet (Baseline).

Figure 5 presents some test results on our test dataset. The results obtained by UNet are poor and noisy. It is easy to identify a pipeline or shadow with a gray value similar to the leakage as the leakage area. Figure 5 shows that the ACPA module greatly reduces the misrecognition. This significant improvement demonstrates that the ACPA module can help ACPA-Net learn strong semantic feature representations effectively. However, as shown in the fourth row of Figure 5, the network is still affected by some noise, which makes the leakage area discontinuous. After adding the deep supervision strategy to the network, the segmentation effect is further improved. The ACPA module is helpful for ACPA-Net to extract effective leakage features. Furthermore, adding extra supervision to the front layer makes ACPA-Net more robust and obtains a more consistent representation inside leakage areas.

### 4.4.3. Comparison with Other Attention Modules

To further verify the effectiveness of the ACPA module in tunnel leakages images segmentation, in this section, the performance of the proposed ACPA module compared with other attention modules is evaluated on our rail tunnel leakage image dataset. We compare it with four other similar lightweight modules, which mainly include the SE module, CBAM, spatial and channel SE (scSE) module [28], and ECA module. The UNet with a deep supervision strategy serves as the baseline and is denoted as "UNet _DS". Then, "+SE", "+CBAM", "+scSE", "+ECA", and "+ACPA" represent the combination of the baseline with attention modules SE, CBAM, scSE, ECA, and ACPA, respectively.

The experimental results are shown in Table 2, where "ΔParam" represents the parameter added to the baseline. It can be seen from Table 2 that compared with the baseline, the network combined with the different attention modules achieves better results. The CBAM, scSE, and SE modules use channel dimensionality reduction to reduce the number of parameters in the attention channel. This strategy leads to an indirect correspondence between channels and weights. It is worth noting that "+ECA" achieves comparable or even better performance than "+SE", "+CBAM", and "+scSE". This result proves that the strategy of avoiding channel dimensionality reduction adopted in the ECA module is necessary to effectively capture cross-channel interactions. "+ACPA" achieves the best performance, with an F1-score, IoU, and Accuracy 0.83%, 1.23%, and 0.19% higher than "+ECA", respectively. Compared with "+SE", "+CBAM", and "+scSE", the parameters of "+ACPA" and "+ECA" are greatly reduced, and "+ACPA" has only 72 parameters more

than "+ECA", which realizes the obvious improvement in F1-score and IoU by increasing fewer parameters.

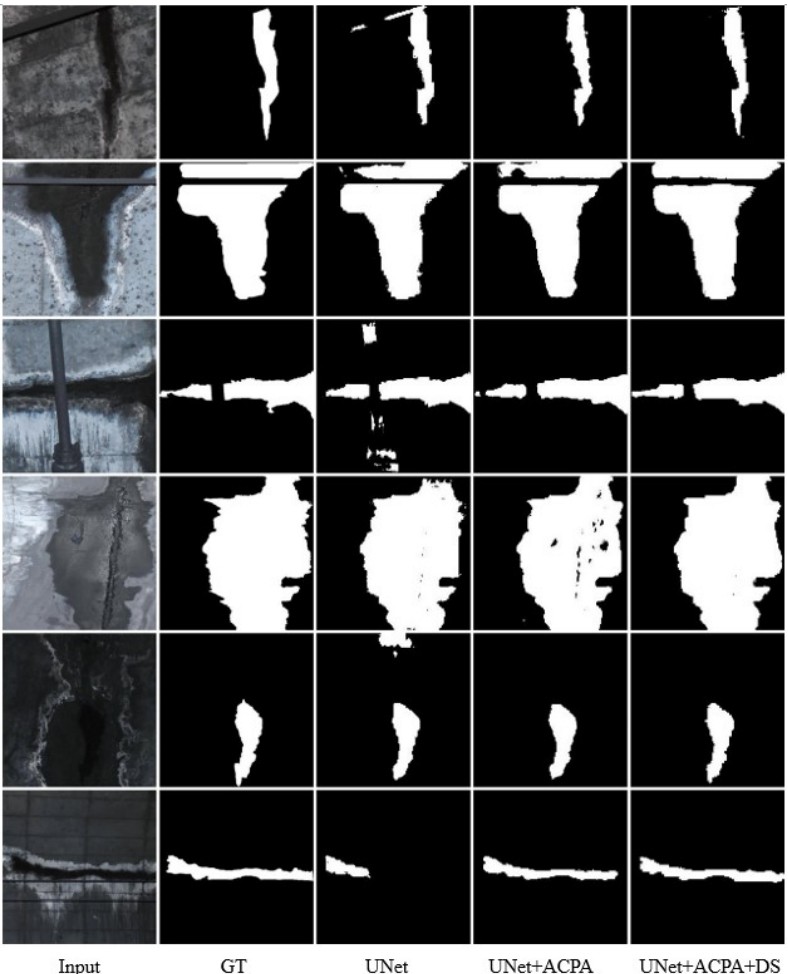

**Figure 5.** Visualization of segmentation results of each component inserted into the baseline.

**Table 2.** Ablation with different attention modules, where UNet+DS serves as the baseline.

| Methods | F1-Score (%) | IoU(%) | Accuracy (%) | Δ Param |
|---|---|---|---|---|
| UNet_DS (Baseline) | 87.81 | 79.20 | 95.77 | – |
| +SE | 89.68 (1.87↑) | 81.76 (2.56↑) | 96.17 (0.40↑) | 13.95 K |
| +CBAM | 89.50 (1.69↑) | 81.69 (2.49↑) | 96.36 (0.59↑) | 30.28 K |
| +scSE | 88.14 (0.33↑) | 79.77 (0.57↑) | 95.64 (0.13↓) | 110.30 K |
| +ECA | 89.92 (2.11↑) | 82.39 (3.19↑) | 96.49 (0.72↑) | **24** |
| +ACPA | **90.75 (2.94↑)** | **83.62 (4.42↑)** | **96.68 (0.91↑)** | 96 |

*4.5. Comparisons with Existing Methods*

4.5.1. Comparisons with Traditional Methods

In this experiment, the proposed method is compared with three representative traditional segmentation methods. The first traditional segmentation method is the Otsu [44] algorithm (OA) based on thresholds. The second one is the region growing algorithm (RGA) based on regional segmentation. The third one is the GrabCut [45] algorithm based on graph theory. If you want to have a deeper understanding of the methods of this paper, please refer to [45]. Table 3 shows the results of three methods above and the proposed method on the test set. Except for Recall, all of the indicators obtained by OA and RGA are far lower than those obtained by the proposed method. Although the OA provides the

highest Recall, reaching 97.67%, the reason for this is that it is difficult for the OA to find an appropriate threshold to automatically separate the leakage areas from the background. It tends to classify the leakage areas and many other background areas into one category, and this causes the Precision to be very small while also obtaining a higher Recall, which is still less than 35%. Therefore, the values of F1-score, IoU, and Accuracy are relatively low. As can be seen from Table 3, GrabCut is superior to other two traditional image segmentation methods, and the proposed method is superior to these three traditional image segmentation methods.

**Table 3.** Comparison with the traditional methods on the test set.

| Methods | F1-Score (%) | Recall (%) | Precision (%) | IoU (%) | Accuracy (%) |
|---------|--------------|------------|---------------|---------|--------------|
| OA | 48.69 | **97.67** | 34.32 | 33.84 | 63.30 |
| RGA | 63.09 | 73.76 | 63.64 | 48.40 | 82.11 |
| GrabCut | 89.29 | 90.68 | 88.75 | 81.34 | 96.25 |
| Ours | **90.75** | 90.47 | **91.56** | **83.62** | **96.68** |

Figure 6 shows some representative images predicted by OA, RGA, GrabCut, and the proposed method. It is obvious that the proposed method provides better results. The segmentation results of traditional methods are easily affected by various factors. It can be seen from the third row in Figure 6 that due to the influence of light reflection, there are many holes in the leakage areas in the results obtained by traditional methods, and only a small part of the leakage area is correctly segmented by the RGA. For the images in the low illumination condition, similar to the image in the last row of Figure 6, RGA and OA algorithms can hardly distinguish the leakage area and the background. GrabCut and ACPA-Net achieve similar performance, but GrabCut requires manual interaction in the process of segmentation; that is, it requires the users to frame the leakage area and mark the incorrectly predicted area appropriately. It exchanges the segmentation result at the cost of manual interaction. In contrast, as a data-driven deep learning method, the proposed method has higher performance and better robustness than the traditional methods without manual intervention.

4.5.2. Comparison with Deep Learning Models

In this experiment, the performance of the proposed method is compared with the current state-of-the-art CNN-based methods and Transformer-based methods. To measure the performance of our method more comprehensively and broadly, we first conduct our experiments on the Crack500 [21] dataset. On this dataset, the CNN-based methods such as PSPNet [46], DeepLabv3+ [47], SFNet [48] are used. These methods adopt ResNet-18 as the backbone network. DeepCrack and an improved FCN [14] method are both disease detection methods for civil engineering structures, which separately study pavement images segmentation and shield tunnels leakage segmentation. In the DeepCrack method, multiscale deep convolution features extracted from hierarchical convolution stages with VGG are fused together to capture the crack region in concrete crack images. The improved FCN method can segment the tunnel leakage intensity image automatically, which is a new method that uses the intensity images of terrestrial mobile LiDAR (Light Detection and Ranging) for automatic leakage detection in shield tunnels based on deep learning. Swin Transformer [49] is a general purpose backbone network that can be used for computer vision tasks. It proposes a hierarchical Transformer; this hierarchical architecture has the flexibility to model at various scales and has linear computational complexity with respect to image size. Due to limited space, only a brief introduction is made here. If you want to have a deeper understanding of this method, please refer to [49]. UpperNet using Swin Transformer as the skeleton is compared with ACPA-Net. In this paper, we name this method UperNet(Swin). In UperNet(Swin), the tiny version (Swin-T) is used. All the state-of-the-art methods are pretrained with the ImageNet dataset to reduce overfitting

and improve the performance of the model to a certain extent. The validation F1-score of different models on different datasets are shown in Figures 7 and 8.

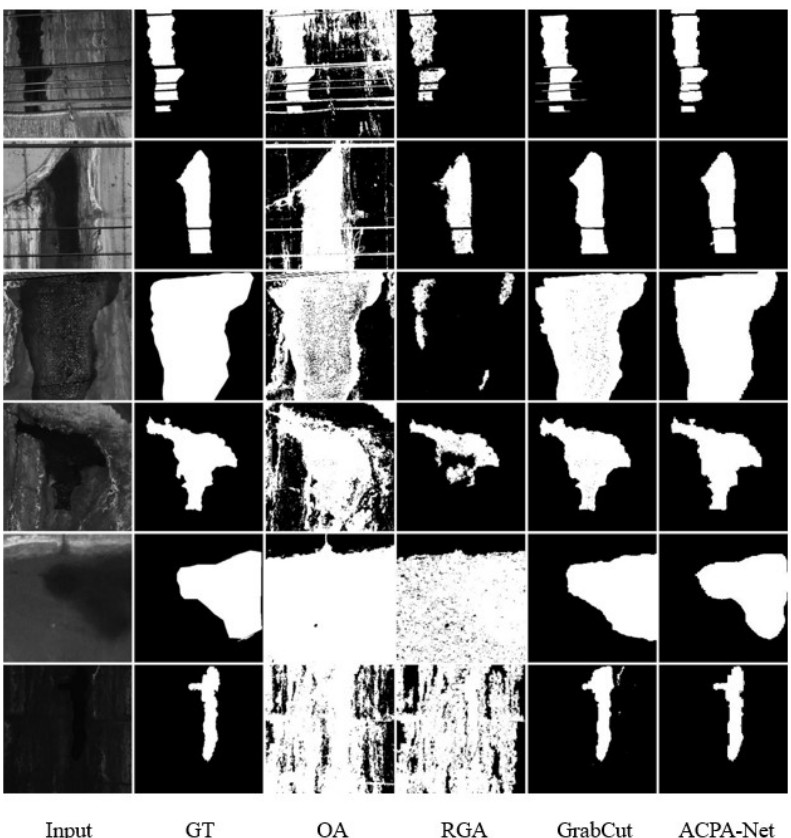

**Figure 6.** Visualization of inferenced results obtained by OA, RGA, GrabCut, and ACPA-Net on test set.

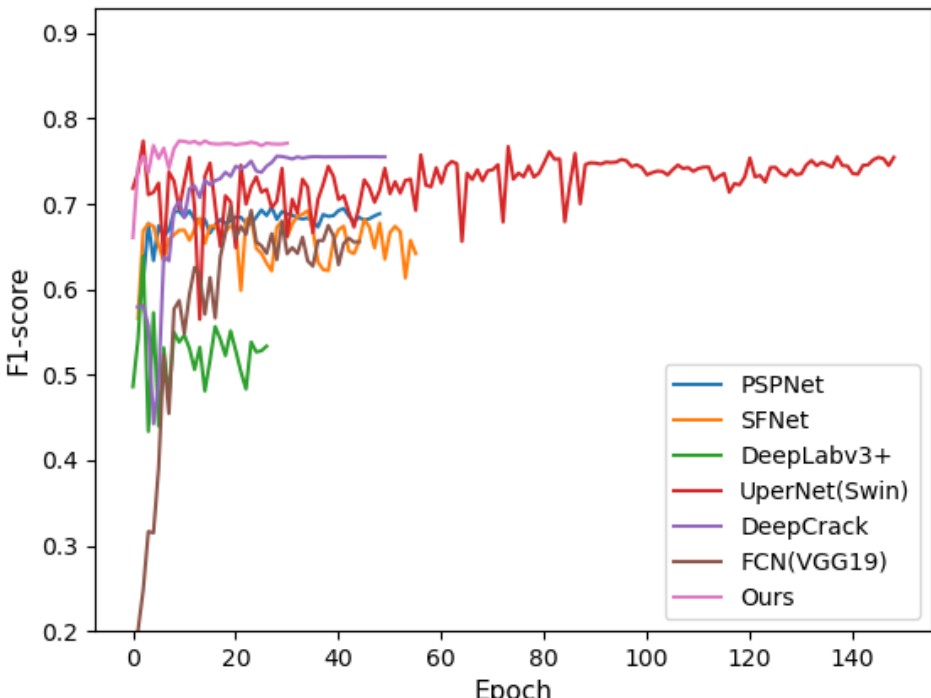

**Figure 7.** Validation F1-score of different models on Crack500 dataset.

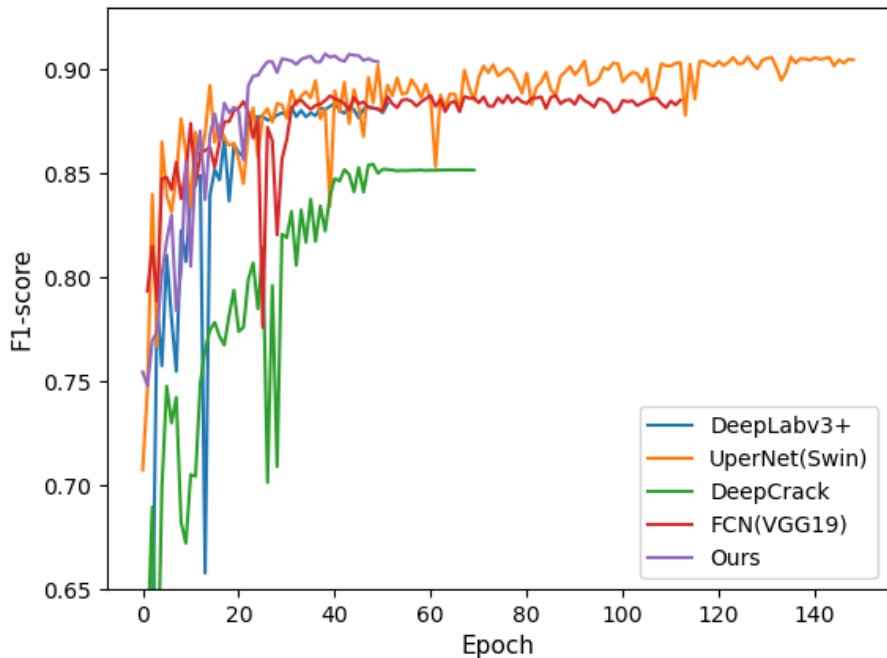

**Figure 8.** Validation F1-score of different models on our tunnel leakage dataset.

Table 4 shows the comparison among the proposed method and the above six methods on the Crack500 [21] dataset. As can be seen from Table 4, except for Precision, ACPA-Net is optimal in other metrics. In terms of F1-score, our method is 2.41% higher than DeepLabv3+, 3.57% higher than DeepCrack and 9.53% higher than FCN(VGG19). As shown in Figure 9, the edge of the crack area in the image predicted by PSPNet is rough, and the crack areas cannot be effectively distinguished. It can be seen that the pyramid pooling module proposed in PSPNet for multi-scale context modeling is not suitable for crack segmentation in complex environments. Atrous convolution is used by DeepLabv3+ to maintain image resolution in the encoder. Then, the ASPP module is used to capture multiscale context information. It can be seen from Figure 9 that the results obtained by DeepLabv3+ are slightly improved in comparison with PSPNet, but the crack areas in the predicted results still have a poor edge. SFNet uses the flow alignment module to align feature maps of two adjacent levels by predicting a flow field. Compared with the previously mentioned PSPNet and DeepLabv3+, SFNet predicted sharper and more accurate crack boundaries, but the boundaries are serrated and not smooth. UperNet(Swin), DeepCrack and FCN(VGG19) fail to completely suppress the background, thereby leading to false detection and missing detection to some extent. For example, the background regions in the middle of the second image are not effectively suppressed, and there is the missing detection phenomenon in the middle area of the image. In addition, as can be seen from the last row of the third image to the ninth image, the cracks predicted by the other six methods are wider than the ground truth and ACPA-Net.

**Table 4.** Comparison with state-of-the-art deep learning models on the Crack500 test set.

| Methods | F1-Score (%) | Recall (%) | Precision (%) | IoU (%) | Accuracy (%) |
|---|---|---|---|---|---|
| PSPNet | 71.69 | 77.27 | 68.80 | 57.40 | 97.18 |
| SFNet | 75.83 | 79.29 | 74.52 | 62.38 | 97.58 |
| DeepLabv3+ | 76.76 | 80.13 | 76.11 | 63.76 | 97.69 |
| UperNet(Swin) | 75.45 | 72.21 | **80.05** | 65.82 | 95.44 |
| DeepCrack | 75.60 | 80.13 | 74.94 | 62.75 | 97.66 |
| FCN(VGG19) | 69.64 | 75.28 | 63.03 | 55.74 | 96.61 |
| Ours | **79.17** | **82.30** | 77.52 | **66.62** | **97.88** |

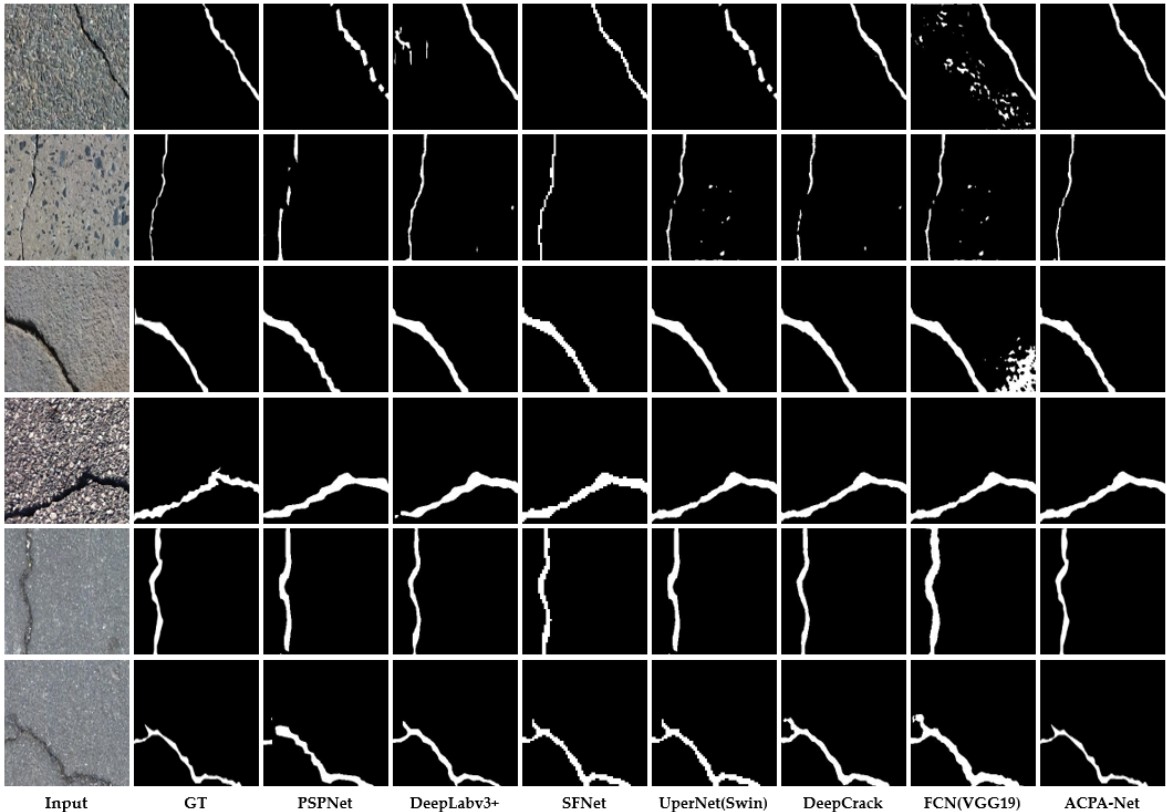

**Figure 9.** Visualization of the inference results obtained by different state-of-the-art deep learning models on the Crack500 test set.

Table 5 shows the comparison among the quantitative results on our tunnel leakage image dataset. As can be seen from Table 5, by using a method where there is a direct correspondence between channels and its corresponding parameters, ACPA-Net shows the best perfomance compared with the results of the other four methods. The parameter of our method is only 4.32 M, which is 167.14 M less than that of FCN(VGG19). DeepCrack as a pavement images segmentation method and FCN(VGG19) as a shield tunnels leakage segmentation method are compared with the proposed method. Due to DeepCrack failing to make full use of multiscale context information, it leads to the accuracy of detecting leakage areas being relatively low. FCN(VGG19) is a FCN-based method, in which the larger ratio upsampling operation makes the predicted segmentation result fuzzier and more insensitive to the details in the image. Therefore, it can be seen that the proposed method has improved 5.58% and 2.01% in F1-score, 5.33% and 7.75% in Recall, and 14.55% and 3.05% in IoU compared with DeepCrack as a pavement images segmentation method and FCN(VGG19) as a shield tunnels leakage segmentation method. We also measure the model by FLOPs (floating point of operations) and FPS (Frames Per Second); FPS is used to evaluate the inference speed of image segmentation, which is a relatively important evaluation metric in engineering. The larger the FPS is, the faster the inference speed. Our method is significantly better than the two methods above in FLOPs and FPS. Compared with DeepLabv3+, although the FPS of this method is higher and FLOPs is smaller, our method has improved in the other six metrics by fusing features from the encoder to supply some detail information, improving 2.38% in F1-score, 3.58% in Precision and 3.86% in IoU. Compared with UperNet(Swin), our method significantly outperforms UperNet(Swin) in Precision and IoU, gains a 2.36% and 0.7% increment in Precision and IoU, respectively, even though UperNet uses the latest Swin Transformer as the skeleton. Furthermore, ACPA-Net is far superior to UperNet(Swin) in FLOPs and FPS; the inference speed of ACPA-Net is 76.66 FPS, which is about twice as fast as UperNet(Swin). The above results

may be due to the fact that UperNet(Swin) is a general method, while our method is a tunnel leakage image segmentation method. In summary, the proposed method achieves the highest inference speed and lowest computational cost except for the DeepLabv3+ method, and it exceeds the other state-of-the-art methods, even the Transformer-based method in the metrics of F1-score, IoU, Precision and Accuracy.

**Table 5.** Comparison with state-of-the-art deep learning models on our tunnel leakage dataset.

| Methods | Param (M) | F1-Score (%) | Recall (%) | Precision (%) | IoU (%) | Accuracy (%) | FLOPs (G) | FPS |
|---|---|---|---|---|---|---|---|---|
| DeepLabv3+ | 12.33 | 88.37 | 89.89 | 87.98 | 79.82 | 95.83 | **4.58** | **124.74** |
| UperNet(Swin) | 59.83 | 90.60 | **92.24** | 89.20 | 82.98 | 96.52 | 59.64 | 38.23 |
| DeepCrack | 30.91 | 85.17 | 85.14 | 82.88 | 69.13 | 93.38 | 136.92 | 5.26 |
| FCN(VGG19) | 171.46 | 88.74 | 82.72 | 87.28 | 80.63 | 92.96 | 36.43 | 34.85 |
| Ours | **4.32** | **90.75** | 90.47 | **91.56** | **83.68** | **96.68** | 10.10 | 76.66 |

Figure 10 demonstrates the leakage segmentation results with the proposed ACPA-Net and the other four methods mentioned above. As can be seen from Figure 10, our method more accurately and completely locates the leakage areas than the CNN-based and Transformer-based methods. For example, in the middle regions of the first image, the other four methods have the missing detection phenomenon, and the DeepCrack method does not effectively suppress the middle background regions of the image. However, with the help of the ACPA module and deep supervision strategy, our method successfully obtains accurate segmentation results. In the upper right and lower left areas of the second image, only our method can locate the leakage regions clearly, accurately, and completely. In addition, the third row of the fifth image and the fifth row of the fourth image fail to effectively suppress the background regions. However, our proposed method exhibits stronger advantages in terms of accurate positioning, background suppression and detection integrity. In general, our method has a more complete structure and clearer boundaries, which benefits from the ACPA module helping ACPA-Net strengthen useful feature channels and suppress useless feature channels. The skip connections between the encoder and decoder can enhance the finer details by recovering local spatial information. A deep supervision strategy can help ACPA-Net effectively improve the discriminativeness and robustness of learned features. The ACPA module uses a local cross-channel interaction strategy without dimensionality reduction, thereby improving accuracy without bringing in a massive quantity of parameters. The atrous convolution in the ACPA module can expand the receptive field, improve ACPA's ability to facilitate more channels' interaction and improve the ability to extract global information. However, the global feature extraction ability of the ACPA module is limited and thus unable to extract the features of leakage areas well. So, in the second row of the last image, a portion of the area in the middle of the image is not predicted as a leakage area but as a lining. The self-attention mechanism of the Transformer exploits global dependencies with a global perceptual field, which has achieved good results on many visual tasks. In the future, we will try to introduce a new Transformer as encoder and decoder to achieve more complete segmentation of the disease areas.

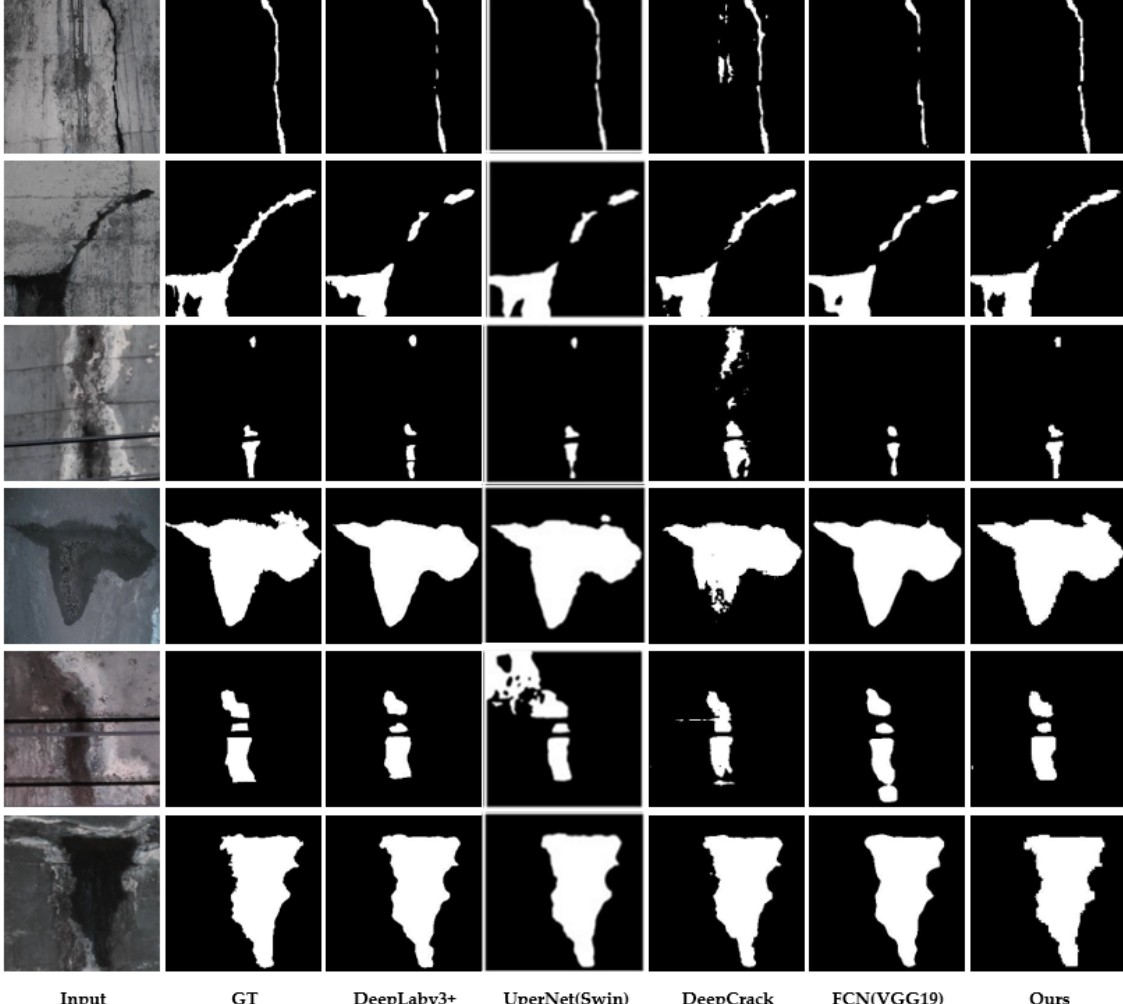

**Figure 10.** Visualization of the inference results obtained by different state-of-the-art deep learning models on our tunnel leakage test dataset.

## 5. Conclusions

In this paper, an atrous channel pyramid attention network (ACPA-Net) is proposed for the segmentation of rail tunnel leakages. Specifically, the ACPA module generates channel attention through multiple parallel 1D dilated convolutions with different atrous rates. With the proposed ACPA module integrated into ACPA-Net, ACPA-Net can learn strong feature representations by considering the relationship among different channels feature. In addition, a deep supervision strategy is introduced to provide direct supervision to the side output. The ablation study demonstrates that the ACPA module can effectively capture long-distance cross-channel interactions with a few parameters, and the deep supervision strategy is effective to improve the discriminativeness and robustness of features learned by ACPA-Net. The final experiment on the railway tunnel leakage image dataset shows that ACPA-Net outperforms three representative traditional image segmentation methods and another four deep learning algorithms. Compared with traditional methods, our method is far superior to traditional methods in most indicators. The inference results show that our method is more robust. Compared with the other four state-of-the-art deep learning methods, the proposed method provides a better representation in leakage regions than other methods under the benchmark of ground truths. The parameter quantity of our method is the least among the other methods.The proposed method achieves the highest inference speed and lowest computational cost except for the DeepLabv3+ method. The proposed method exceeds the other state-of-the-art methods, even the Transformer-based

method in the metrics of F1-score, IoU, Precision and Accuracy. It can be said that the proposed method has the best comprehensive performance in terms of parameter quantity, calculation cost, inference speed and segmentation result. In future studies, more work should be conducted to reduce the computational cost, accelerate the inference speed and improve the robustness of the proposed model. In addition, since our method is a tunnel leakage image segmentation method, we can add special defect segmentation, such as tunnel lining cracks segmentation. Although ACPA-Net has the lowest number of parameters among all comparison methods, the information obtained by long-distance convolution has no correlation and lacks interdependence. Therefore, the prediction results in some images are not perfect. However, Transformer methods have certain advantages in global modeling. In future research, we will try to introduce new transformer methods to achieve more complete segmentation of the disease areas.

**Author Contributions:** Conceptualization, P.G., Z.T. and J.L.; methodology, P.G., Z.T., J.L. and J.B.; software, Z.T. and J.L.; validation, J.B.; investigation, Z.T., T.W., F.L. and J.B.; data curation, P.G., Z.T. and J.L.; writing—original draft preparation, P.G., Z.T., J.L., T.W. and F.L.; writing—review and editing, P.G., Z.T., J.L., T.W. and F.L.; supervision, P.G.; funding acquisition, T.W. and F.L. All authors have read and agreed to the published version of the manuscript.

**Funding:** This research was funded by China Postdoctoral Science Foundation grant number 2022TQ0218 and funded by The Science and Technology Project of Hebei Education Department grant number ZD2022098.

**Institutional Review Board Statement:** Not applicable.

**Informed Consent Statement:** Not applicable.

**Data Availability Statement:** The code and weights on Crack500 dataset of our ACPA-Net can be found from the links of https://github.com/sineagles/ACPA and https://www.researchgate.net/publication/366823076_ACPA-Net.

**Conflicts of Interest:** The authors declare no conflict of interest.

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
