# Peer review of "ACPA-Net: Atrous Channel Pyramid Attention Network for Segmentation of Leakage in Rail Tunnel Linings"

_electronics, doi:10.3390/electronics12020255_

Round 1

Reviewer 1 Report

Dear Authors,

thank you for the interesting contribution. The paper is well organized and presents quality scientific work.

The main point of criticism is the missing delimitation to approaches having a similar performance: namely, GrabCut, SegFormer and Swin Transformer.

1. What is computational time need by your and named methods? This question is crucial from my point of view.

2. How well is your method applicably to other segmentations tasks? This question should be answered at least on a theoretical level. Even better supported by some experiments.

Kind regards

Reviewer 2 Report

The authors address the problem of leakage detection in rail tunnels. To this aim, they propose to improve DL semantic segmentation network with two main techniques: cross-channel attention module and multi-level prediction through a side network. More than this, they collect a manually labelled dataset for evaluating their and multiple other state-of-the-art approaches.

The contribution of this paper is of great interest not only for the specific application problem at hand, which the authors are still skillfully able to use for motivating their work, but also for the broader semantic segmentation topic. Furthermore, the dataset itself may be listed among the contributions if they could release a documented version open-access to the public.

On the writing side, there is no point to be made by my side. The text is precise in all its parts, and the method is well illustrated graphically. The experimental section provides all the ablations studies a reader would expect.

Regarding the DS method, which provides a slight increase in performance, I would be curious to see if downsampling the labels would be more effective. This is usually the preferred approach in many tasks (e.g., depth estimation or optical flow).

Reviewer 3 Report

1.      Key ideas, experiments, and their significance

The authors proposed a binary semantic segmentation solution using (1) a U-shaped encoder-decoder network architecture, (2) skip connections are used in ACPA-Net to combine the high-resolution feature maps coming from the encoder with the upsampled output, (3) atrous channel pyramid attention (ACPA) is adopted, (4) a side network is added to the decoder to further supervise it. Overall, the authors’ contributions include: (1) lightweight channel attention ACPC module to capture longer distance channel interaction of feature extraction, and (2) an overall ACPA-Net with good results in terms of accuracy and size trade-offs. The experiments are clear, but performance increases are specific to the domain of detecting rail tunnel leakage.

2. Strengths of the paper

-          The paper is decently written and well formatted.

-          The authors proposed a decent engineering approach to improving Unet-based frameworks of semantic segmentation.

-          Detailed ablation and general experiments are provided, all with clear descriptions.

3. Weaknesses of the paper and comments

-          Limited dataset of images in training is 1100 and that in test dataset is 270. The authors should either extend it, or propose ACPC as a general component for Video Object Segmentation solutions to take into account more data, thereby demonstrating the true superiority.

-          Insufficient proof for superiority of ACPA component. Because the results are surprisingly even better than existing state-of-the-art approaches like SwinTransformer, SegFormer, the authors ought to investigate these findings in-depth (i.e. specifically why is it better, or competitive against these popular approaches). In particular, they may discuss whether these increases are because of specific characteristics of the tunnel leakage dataset, or could be extended to general cases, or weaknesses within SwinTransformer itself.

-          In the related works, authors consider only object detection methods, instead of image segmentation approaches, such as those discussed in [1]. The authors should also extend it to accommodate the methods that they compared in the experiemnts.

-          Limited scientific and technical contribution because of only one major contribution, but margin of increase is small and implementation (code) details are hidden. Hence, both theorectical and practical aspects of novelty are limited. The authors should discuss carefully why ACPA can work well with detecting tunnel leakages, what are the theoretical/heuristic bases for it, and if ACPA-Net has any weaknesses that can be investigated for future improvements.

[1] S. Minaee, Y. Boykov, F. Porikli, A. Plaza, N. Kehtarnavaz and D. Terzopoulos, "Image Segmentation Using Deep Learning: A Survey," in IEEE Transactions on Pattern Analysis and Machine Intelligence, vol. 44, no. 7, pp. 3523-3542, 1 July 2022, doi: 10.1109/TPAMI.2021.3059968.

Round 2

Reviewer 1 Report

Dear Authors,

thank you for the response and clarification.

The manuscript can be accepted as it is.

Kind regards

Artem Leichter

Author Response

Thank the reviewer for your comments on our paper.

Reviewer 3 Report

The authors proposed a binary semantic segmentation solution using (1) a U-shaped encoder-decoder network architecture, (2) skip connections are used in ACPA-Net to combine the high-resolution feature maps coming from the encoder with the upsampled output, (3) atrous channel pyramid attention (ACPA) is adopted, (4) a side network is added to the decoder to further supervise it. Overall, the authors’ contributions include: (1) lightweight channel attention ACPC module to capture longer distance channel interaction of feature extraction, and (2) an overall ACPA-Net with good results in terms of accuracy and size trade-offs. The descriptions for methodology and experiment are sufficiently conveyed, but the authors need to clarify a few things regarding their motivation, and the highlight of their approach compared to existing works.

The paper is decently written and well formatted. 

The authors developed ACPA, which addresses the limited interaction range between channels of k adjacent channels in ECA. With ACPA, the authors considered a wider range of cross-channel interactions with multiple parallel 1D dilated convolutions. The authors’ hypothesis is demonstrated empirically with improve accuracies on the leakage segmentation problem.

Detailed ablation and general experiments are provided, all with clear descriptions.

In the introduction and related works, it is not yet clear about the authors’ motivation. In particular, the authors did not provide existing sources to support why interdependencies between feature channels of neural networks are important. The authors are apparently not the first to propose the idea of channel-wise interdependencies, so it is best that the concepts are laid out carefully. In particular, please carefully cite and discuss what other methods have investigated the importance of channel-wise interdependencies (e.g. ECA), and provide reasoning and comparison of how they work compared the the proposed ACPA.

In the related works, although the authors have added some discussions about semantic segmentation algorithms (per my previous comments), they have yet highlighted the significance of the ACPA-Net framework when compared to others in the field of leakage detection. In particular, please discuss why the authors’ proposed framework’s design is superior to those of existing frameworks in the field. The authors ought to discuss it against some existing methods [1], [2], [3], [4].

In the experiments, the authors have forgotten to cite the paper for the Crack500 dataset. Please credit the dataset’s authors.  

Theoretical aspects of novelty are limited. The authors should discuss more in the experimental perspectives:   What are the difficult cases that not even ACPA-Net can address in the datasets (i.e. why is the accuracy 90% on the private dataset and 70% on Crack500, but not perfect). This serves future research in the field. 

[1] H. Li, J. Zong, J. Nie, Z. Wu and H. Han, "Pavement Crack Detection Algorithm Based on Densely Connected and Deeply Supervised Network," in IEEE Access, vol. 9, pp. 11835-11842, 2021, doi: 10.1109/ACCESS.2021.3050401.

[2] Yang, F., Lei Zhang, Sijia Yu, Danil V. Prokhorov, Xue Mei and Haibin Ling. “Feature Pyramid and Hierarchical Boosting Network for Pavement Crack Detection.” IEEE Transactions on Intelligent Transportation Systems 21 (2020): 1525-1535.

[3] Hong, Youn-Chan, Sung-Jin Lee and Seok Bong Yoo. “AugMoCrack: Augmented morphological attention network for weakly supervised crack detection.” Electronics Letters (2022): n. pag.

[4] C. Han, T. Ma, J. Huyan, X. Huang and Y. Zhang, "CrackW-Net: A Novel Pavement Crack Image Segmentation Convolutional Neural Network," in IEEE Transactions on Intelligent Transportation Systems, vol. 23, no. 11, pp. 22135-22144, Nov. 2022, doi: 10.1109/TITS.2021.3095507.
